# Healthcare professionals' experiences of job satisfaction when providing person-centred care: a systematic review of qualitative studies

Kristoffer Gustavsson ![ORCID],[1] Cornelia van Diepen ![ORCID],[1,2] Andreas Fors ![ORCID],[1,3,4] Malin Axelsson,[5] Monica Bertilsson,[6] Gunnel Hensing[6]

For numbered affiliations see end of article.

**Correspondence to**
Dr Andreas Fors;
andreas.fors@gu.se

## ABSTRACT

**Objectives** This qualitative systematic review aimed to explore and synthesise healthcare professionals' (HCPs) experiences of job satisfaction when providing person-centred care (PCC) in healthcare settings in Europe.

**Method** This systematic review of qualitative studies was followed by a thematic synthesis applying an inductive approach. Studies concerning HCPs and different levels of healthcare in Europe were eligible for inclusion. The CINAHL, PubMed and Scopus databases were searched. Study titles, abstracts and full texts were screened for relevance. Included studies were assessed for methodological quality using a quality appraisal checklist. Data were extracted and synthesised via thematic synthesis, generating analytical themes.

**Results** Seventeen studies were included in the final thematic synthesis, and eight analytical themes were derived. Most studies were conducted in Sweden and the UK and were performed in hospitals, nursing homes, elderly care and primary care. Thirteen of these studies were qualitative and four used a mixed-method design in which the qualitative part was used for analysis. HCPs experienced challenges adapting to a new remoulded professional role and felt torn and inadequate due to ambiguities between organisational structures, task-oriented care and PCC. Improved job satisfaction was experienced when providing PCC in line with ethical expectations, patients and colleagues expressed appreciation and team collaboration improved, while learning new skills generated motivation.

**Conclusion** This systematic review found varied experiences among HCPs. Notably, the new professional role was experienced to entail disorientation and uncertainty; importantly, it also entailed experiences of job satisfaction such as meaningfulness, an improved relationship between HCPs and patients, appreciation and collaboration. To facilitate PCC implementation, healthcare organisations should focus on supporting HCPs through collaborational structures, and resources such as time, space and staffing.

**PROSPERO registration number** CRD42022304732.

## INTRODUCTION

In Europe, healthcare professionals (HCPs) are at higher risk of exposure to psychosocial

### STRENGTHS AND LIMITATIONS OF THIS STUDY

⇒ This is the first systematic review synthesising qualitative research on healthcare professionals' experiences of job satisfaction when providing person-centred care across different healthcare settings in Europe.
⇒ The methodology was thoroughly described, increasing replicability and reliability, and the researchers involved in this review had different professional backgrounds, that is, nursing, sociology and social medicine.
⇒ The search strategy was extensive, included a broad range of terms and covered several databases; a search specialist was consulted in the process.
⇒ The search was limited to studies written in English and published in or since 2010, so some potentially relevant studies might have been missed.
⇒ The included studies had different qualitative methodologies and theoretical approaches, which could affect interpretations and comparisons.

risk factors from their work environment than are several other occupations,[1] which can result in adverse health impacts on the individual, such as stress, fatigue, burnout and physical and emotional demands.[2] The working environment affects job satisfaction, which is a mental state of satisfaction deriving from employees' assessment of their work situation.[3] It includes factors such as accomplishment, praise, social relations, advancement opportunities and undertaking tasks in line with personal ethical values.[4 5] Low job control can decrease health and job satisfaction,[6] and high stress combined with decreased job satisfaction is associated with high turnover rates in healthcare.[7] With demands on European healthcare systems gradually increasing,[8] recruiting and retaining educated HCPs is difficult, threatening the sustainability of healthcare.[9] Moreover, an undersupported workplace with many stressors can lead to

decreased quality of care for patients and increased emotional demands on HCPs.[10]

Simultaneously, European healthcare has been evolving to involve more patient participation.[11] Person-centred care (PCC) is a model of care deriving from an ethical standpoint in which patients' capabilities and needs are emphasised, and care is provided through collaboration between professionals, patients and family.[12 13] Providing PCC involves some key principles, such as: initiating and focusing partnership by carefully listening to the patient's narrative, with the patient seen as an expert on her/his health; and co-creating a health plan covering the patient's experiences, capabilities, goals, expectations and social support resources along with medical status, and modifying the health plan according to the circumstances.[12 14 15] PCC communication involves both verbal (eg, open-ended questions, reflections and summaries) and non-verbal communication (eg, displaying welcoming and respectful body language).[14 16]

Working with PCC based on the patient's narrative helps professionals realise their expected ethical standards and provide high-quality care.[17] Working in a more person-centred way requires conducive organisational structure and support, leadership and training and constructive collaboration within the interprofessional team.[18 19] Several concepts have been used to describe care centred around the patient and her/his family, such as patient-centred,[20 21] relationship-centred,[22] family-centred,[23] and individualised care,[24] and varied settings require specific approaches of 'centredness' on the patient and family.[25 26] PCC entails building a relationship in which patients are involved as active partners in their care and treatment and, more distinctly, emphasises a shift away from a model of care in which patients are often regarded as passive care recipients.[13 15]

In European healthcare, ethical stress and lack of resources are issues affecting HCPs' ability to offer the desired care of the required quality.[27] PCC has been proposed as a model of care with potential to help solve some of these problems, although evidence regarding PCC and its impact on HCP outcomes remains unclear.[28] PCC implementation being increasingly integrated into healthcare systems internationally will expose more HCPs to this model of care. Findings from this systematic review can help enhance the understanding of how HCPs' job satisfaction relates to PCC, and lead to further research and new relevant policies to improve working conditions in healthcare.

Research has primarily investigated the patient outcomes of PCC, showing, for example, increased self-efficacy, improved satisfaction with care, improved symptom control and clinical outcomes characterised by shorter hospital stays and cost savings.[29] Six reviews of aspects of HCP outcomes and PCC have been identified,[30–35] which presented mixed findings. These reviews focused on specific professions, such as registered nurses (RNs), nurse aides, direct-care workers and caregivers in residential care or nursing homes,[30–32 34 35] or included only quantitative studies.[30 31 33–35] No reviews have explored qualitative research on job satisfaction from providing PCC across healthcare professions and settings in Europe; this systematic review accordingly aimed to fill this research gap.

## AIM
This qualitative systematic review aimed to explore and synthesise HCPs' experiences of job satisfaction when providing PCC in healthcare settings in Europe.

## METHODS
### Design
A systematic review of qualitative studies was conducted. The research question was 'How do healthcare professionals in Europe, when providing PCC, experience their job satisfaction?' A protocol following the Preferred Reporting Items for Systematic Reviews and Meta-Analyses (PRISMA)-Protocols checklist was written in preparation for the systematic review and registered in PROSPERO on 23 January 2022.[36] A systematic search was executed on 21 December 2021 in the electronic databases CINAHL, PubMed and Scopus, covering the main research areas relevant to this review. An updated search, using the same search strategy for studies published in the period between December 2021 and March 2023 was conducted in March 2023. Thematic synthesis with an inductive approach was used to analyse data obtained from the relevant studies.[37] The framework for Enhancing Transparency in Reporting the Synthesis of Qualitative Research[38] was used for reporting the synthesis (see online supplemental file 1).

### Eligibility criteria
The eligibility criteria for the literature search in this systematic review are presented in table 1, and were based on the Population, Exposure and Outcome (PEO) framework.[39] The inclusion criterion for the population was HCPs, for exposure it was PCC and for the outcome it was experiences of job satisfaction. Additionally, studies from all healthcare settings were acceptable, provided they were conducted in European countries with a healthcare

**Table 1**  Eligibility criteria

| Inclusion criteria | Exclusion criteria |
|---|---|
| ► Healthcare professionals (Population).<br>► Person-centred care (Exposure).<br>► Experiences of job satisfaction (Outcome).<br>► All healthcare settings.<br>► Studies conducted in European countries with identified healthcare systems according to Rosengren *et al*.[40]<br>► Qualitative, peer-reviewed and ethically approved studies, written in English.<br>► Published in or since 2010. | ► Quantitative studies.<br>► Reviews. |

system that could be identified according to Rosengren *et al*.[40] Studies had to be qualitative, peer-reviewed, ethically approved, written in English and published in or since 2010. The included qualitative studies also include the qualitative, but no quantitative, parts of mixed-method studies. The exclusion criteria were quantitative studies and reviews.

## Search strategy

The search strategy was created by the research group with assistance from a librarian at the Biomedical Library at the University of Gothenburg. It comprised keywords and Medical Subject Heading terms hierarchically structured to facilitate the inclusion of broader and more precise medical and health-related terms.[41] In accordance with the PEO framework, relevant search terms associated with HCPs, PCC and job satisfaction were defined and put in their associated search blocks.[39] Within each block, the search terms were connected using the Boolean operator 'OR', and the blocks were connected with the Boolean operator 'AND'.[42] Along with the PEO blocks, two additional blocks were added. The first additional search block, 'countries', consisted of the European countries eligible for inclusion, similar to Rosengren *et al*[40] and their mapping of European countries having identifiable and eligible healthcare systems. Rosengren *et al* found 23 eligible European countries and identified three types of healthcare systems: Beveridge (n=12), Bismarck (n=10) and Out of Pocket (n=1).[40] The second additional block was 'study design', which comprised different search terms for qualitative study designs and mixed-method studies (eg, interviews and focus groups) to ensure that the search would identify all relevant studies of qualitative design. The search strategy is presented in online supplemental file 2.

## Selection process

The selection of studies was conducted between 25 January 2022 and 11 March 2022. The 3754 records from the database searches were imported into EndNote for reference management and into Rayyan software for screening. The first (KG) and second author (CvD) screened titles and abstracts blinded and independently for relevance in Rayyan using a screening tool (see online supplemental file 3).[43] The 'detect duplicates' option available in both programmes was used to remove 872 duplicates.[44 45] The remaining 2882 studies were screened for title and abstract, and 2807 of those were excluded. Out of the 75 studies, 1 study was not retrieved in full text. In the second step of the selection process, the remaining 74 studies were downloaded and shared in an EndNote library. The studies were read in full text for relevance against the inclusion criteria by KG and CvD independently in EndNote. All full-text studies were discussed in a meeting among three authors (KG, CvD, and AF) until consensus was reached. Fifty-six studies were excluded due to incorrect outcome, exposure or design, or by being conducted in the wrong location, or not being written in English.

Eighteen studies were considered relevant to the review. After closer reading, one study was interpreted as related to working during the COVID-19 pandemic rather than to person-centred work; therefore, it was excluded at this stage after discussions. The selection process is presented in a PRISMA flow diagram (figure 1).[46]

For the updated literature search conducted in March 2023, the same electronic databases and search strategy were used, but the year of publication was limited to 2021–2023. The records were imported into EndNote and screened for relevance by the first author (KG). Two relevant studies were identified, which will be incorporated in the discussion section.

## Quality assessment

The Swedish Agency for Health Technology Assessment and Assessment for Social Services (SBU) checklist for assessing methodological quality was used to assess the included studies after they were read in full text.[47] This checklist has five domains (ie, theoretical framework, sampling, data collection, analysis and researcher role) and 13 items integrated among the five domains (see online supplemental file 4).[48]

The studies were initially assessed by the first author (KG), and then independently assessed by the last author (GH), who is a senior researcher and professor. Both KG and GH considered omitting studies indicating low methodological quality. All studies considered relevant were assessed as having moderate or high quality by both assessors. Items in the five domains were related to methodological choices, such as if the aim was coherent with regard to the theoretical stance, if the recruitment was suitable and well conducted or if the method of analysis was appropriate and carried out properly. The items could be answered with 'yes', 'no' or 'unclear'. The combined item answers regarding whether it had serious methodological concerns affecting the reliability of the study resulted in a domain grading of 'yes' 'no' or 'unclear' (see online supplemental file 4). The gradings from the five domains were then combined and given an overall rating of the study as follows: no or few concerns=high quality; moderate concerns=moderate quality; and high concerns=low quality, in line with the recommendations of SBU.[49] The studies' ratings varied between moderate (n=12) and high (n=5) quality. The most common deficiencies were missing descriptions of the researcher's role and preunderstanding, and how it might have affected the study results. No studies were assessed as having low methodological quality, so no studies were excluded due to the quality assessment. The quality assessments are presented in online supplemental file 5.

## Data extraction and synthesis

The qualitative findings extracted from the included studies were analysed via thematic synthesis. The synthesis followed the three stages described by Thomas and Harden.[37] The synthesis was conducted by KG supported

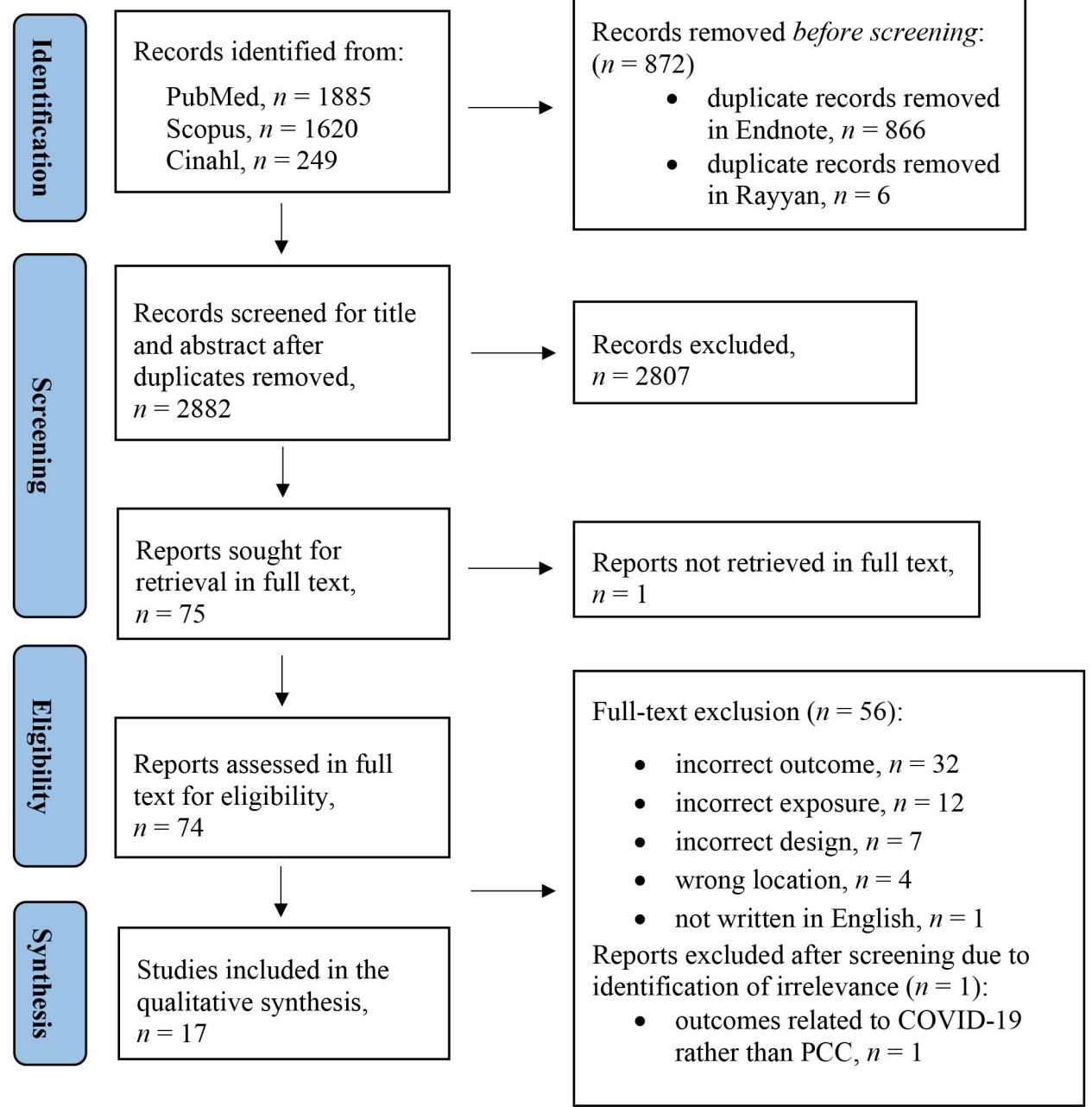

**Figure 1** The selection of studies presented in a Preferred Reporting Items for Systematic Reviews and Meta-Analyses flow diagram.[46] PCC, person-centred care.

by GH. All authors then reviewed and discussed the synthesis and themes.

The first stage of the thematic synthesis was coding, which was conducted using NVivo software, in which the text was coded line by line to identify passages of text from the studies in line with the aim and research question of the review. Each text passage received a code describing its meaning. The coding was continuously scrutinised: it was revised with further coding if consistency was lacking, or progressed without revision if no issues emerged.[37]

The second stage involved inductively developing descriptive themes from the data. In this stage, a descriptive theme was formulated to cover the first code from the first study; either the following codes from the same study were grouped under that theme, if the codes were related in meaning and concept, or new descriptive themes were formulated when codes with new thematic concepts emerged. Codes from subsequent studies were then assigned to existing descriptive themes or new themes were formulated, if needed.[37]

The third stage of the process 'went beyond' the content of the primary studies and created a new understanding of the topic. New knowledge was created by using the descriptive themes generated and connecting them to the research question to produce analytical themes. The analytical themes were discussed by KG and GH, and new themes were created if necessary and scrutinised again

until they were considered to sufficiently describe the descriptive themes.[37]

## Patient and public involvement

This study was conducted without patient or public involvement.

## RESULTS

### Characteristics of included studies

The study aimed to explore HCPs' experiences of job satisfaction when providing PCC. The characteristics of the 17 included studies are presented in online supplemental file 6. Seven studies were conducted in Sweden,[50–56] four in the UK,[57–60] two in the Netherlands,[61 62] one in Ireland,[63] one in Portugal,[64] one in Austria and Germany[65] and one in Sweden, Norway and Australia.[66]

Most of the studies were qualitative (n=13),[50–55 57–59 62 63 65 66] while four[56 60 61 64] used mixed methods. Eight studies[52 54 56–58 62 63 66] used individual interviews, three[61 64 65] used focus groups and six[50 51 53 55 59 60] combined individual and focus-group interviews, of which two[51 59] also used dyadic interviews.

The total number of participants in the included studies was 459 (range 6[54]–97[51]). Thirteen studies[50–57 61–64 66] reported the gender of participants, who were predominantly women (n=351; 90%). The professions represented were RNs and specialist nurses (n=211),[50–54 56 58 59 62 63 65 66] assistant nurses, students, enrolled nurses, caregivers, direct-care workers, front-line staff (n=189),[51 53 57 58 60 61 64 66] physiotherapists (n=23),[51 55 56 66] occupational therapists (n=15),[51 56 66] physicians (n=10),[51] professional support workers and social workers (n=8)[58 62] and 'other', with the profession not being specified (n=1).[51] Nine studies were conducted in hospitals,[51–55 58 59 63 65] six in elderly-care, residential-care or nursing homes[56 57 60 61 64 66] and two in primary-care or general practitioner practices (an organisation of one or more general practitioners).[50 62]

### Synthesis

Twenty-four descriptive themes were derived from the codes, and eight analytical themes from the descriptive themes (table 2). The analytical themes were generated by going beyond the content of the primary studies, representing interpretations of the included studies as a whole.[37]

### Analytical themes

#### HCPs feeling torn and inadequate

Working in line with PCC was experienced as challenging and could make HCPs feel torn and inadequate. The reason for this was described as a conflict between the desire to perform personalised care fulfilling ethical

**Table 2** Themes from the qualitative synthesis

| Analytical themes | Descriptive themes | Studies included in themes |
|---|---|---|
| HCPs feeling torn and inadequate | Conflict between organisational structures and PCC. Conflict between disease-oriented task care and PCC. Feelings of inadequacy and guilt. | Coyne,[63] Fridberg et al,[51] Kadri et al,[57] Kjörnsberg et al,[54] Pinkert et al,[65] Uittenbroek et al.[62] |
| PCC demanding for HCPs | High workloads hamper the work with PCC. PCC not affecting or is negative for workload. Stressful due to lack of resources. | Boersma et al,[61] Boström et al,[50] Coyne,[63] Fridberg et al,[51] Kadri et al,[57] Karlsson et al,[52] Kjörnsberg et al,[54] Ross et al,[58] Sjöberg and Forsner,[55] Uittenbroek et al.[62] |
| Remoulded professional role | Disorientation and uncertainties with new approach and routines. Loss of control. Patients having considerable influence. | Boström et al,[50] Fridberg et al,[51] Kadri et al,[57] Sjöberg and Forsner.[55] |
| Providing PCC meaningful for HCPs | Improved job satisfaction. Meaningfulness—making a difference and doing the 'little extra'. In line with ethical expectations. Improved relationship between HCPs and patients. Self-care for improved care. | Barbosa et al,[64] Boersma et al,[61] Fridberg et al,[51] Karlsson et al,[52] Kjörnsberg et al,[54] Nilsson et al,[53] Pinkert et al,[65] Ross et al,[58] Vassbø et al,[66] Öhman et al.[56] |
| HCPs feeling appreciated | Gratitude from patients makes working with PCC easier. Feeling valued and appreciated by colleagues. | Barbosa et al,[64] Boersma et al,[61] Nilsson et al,[53] Uittenbroek et al,[62] Vassbø et al,[66] Öhman et al.[56] |
| Improved team collaboration | Enhanced collaboration improves satisfaction and working context. Social support reduces stress and frustration. | Boersma et al,[61] Kirkley et al,[60] Nilsson et al,[53] Uittenbroek et al,[62] Vassbø et al.[66] |
| Workload and stress reduction | Improves flow and workload. Reduced stress at work. Sense of increased control and reduced anxiety. | Barbosa et al,[64] Fridberg et al,[51] Kjörnsberg et al,[54] Vassbø et al.[66] |
| Increased personal motivation and commitment | Feeling joy and energy. Thriving at work. Willingness and motivation to perform 'ideal' PCC. | Barbosa et al,[64] Fridberg et al,[51] Kadri et al,[57] Karlsson et al,[52] Vassbø et al,[66] Walker and Deacon.[59] |

HCP, healthcare professional; PCC, person-centred care.

and professional standards, organisational deficiencies in support and negotiation and an organisational focus on effectiveness.[54 57 62 63 65] A perceived conflict between disease-oriented task care and PCC was described, with HCPs initially uncertain whether, for example, listening attentively was regarded as a task.[62] HCPs who experienced difficulties balancing PCC with the more task-oriented disease care perceived that they were blamed by others, resulting in feelings of guilt and inadequacy.[51 54]

### PCC demanding for HCPs

HCPs described how spending time with patients when providing PCC could be demanding and be hampered by high workloads in the organisation.[50 52 55 63] HCPs found that PCC resulted in the same or even a greater workload than before.[51 54 58 62] HCPs also perceived that it could be stressful to work with PCC due to pressures from insufficient resources in terms of time, space and staffing.[51 57 63] Some HCPs found that PCC did not influence their job satisfaction, especially when dealing with difficult behaviours of patients, such as when they did not respond well.[61]

### Remoulded professional role

HCPs experienced a remoulded professional role when providing PCC. This could result in disorientation and uncertainty due to new structures, routines and adaptation to new ways of working.[50 55] They perceived their expertise to be compromised and experienced a loss of control, with patients now having more influence over their own care.[55 57] Moreover, HCPs experienced doubt concerning parts of their new role, which was more conversation based than before and felt less valid than the usual, more hands-on approach.[55] Providing PCC could be frustrating with patients who were less communicative and more reserved[55] and in situations in which HCPs did not agree with patients' wishes regarding their care.[51]

### Providing PCC meaningful for HCPs

When HCPs could provide PCC in line with ethical expectations and could collaborate with patients, they experienced increased meaningfulness and job satisfaction.[52 53 56 58 65 66] PCC provision was aligned with HCPs' standards and expectations regarding ethics and routines, such as providing equitable and inclusive care, discussing goals and writing health plans with patients.[51] Spending more time with patients improved the relationship between HCPs and patients.[51 52 54 58 61 66] Additionally, HCPs described becoming more aware of the importance of their personal well-being when providing PCC, through its connection to patient well-being. HCPs taking better care of themselves could enable care that both patients and HCPs were content with.[64]

### HCPs feeling appreciated

Although some HCPs found that PCC did not influence their job satisfaction when dealing with patients presenting difficult behaviours, others found that providing PCC instead helped in the care of these patients, causing them to show appreciation.[61] Patient expressions of gratification with the care were experienced as leading to increased job satisfaction and enhanced energy.[56 61 66] PCC was also described as a practice leading to appreciation from colleagues, as colleagues could understand and acknowledge one another more through collaboration and interaction, increasing satisfaction and meaningfulness at work.[53 62 64 66]

### Improved team collaboration

The enhanced awareness, learning and collaboration with colleagues and organisations derived from PCC was described as improving job satisfaction and the work environment.[61 62 66] HCPs valued the social support that emerged from working with PCC, fostering a supportive culture in which colleagues shared experiences and feelings, and recognised and relieved each other; this was experienced as increasing job satisfaction and reducing stress and frustration.[53 60 66]

### Workload and stress reduction

HCPs who implemented PCC experienced a better workflow with new routines[51] and a calmer work environment.[66] Lower workload and reduced stress were experienced from working with PCC and being responsible for fewer patients.[54] Moreover, HCPs described increased control arising from enhanced independence in organising the work.[66] With PCC, HCPs could better deal with situations formerly experienced as stressful, which was described as reducing anxiety.[64]

### Increased personal motivation and commitment

Motivation[57 66] and commitment[52 57 59 64] to continue providing PCC increased. The HCPs experienced enhanced joy from the improved connections with patients[51] and increased energy when they could meet patients' needs.[66] Moreover, HCPs working jointly with PCC, helping one another and discussing the care, generated motivation and an increased striving to improve their current skills.[66]

## DISCUSSION

This systematic review identified, assessed and synthesised qualitative studies to explore HCPs' experiences of providing PCC by focusing on job satisfaction. In the updated literature search, two additional relevant qualitative studies by Allerby et al,[67] and Petersson et al[68] were identified. Findings from these studies were in line with findings from this systematic review and will be incorporated into the following discussion.

An interesting finding derived from the synthesis concerned the remoulded professional role. A study by Boström et al[69] of RNs' experiences of providing PCC over the phone highlighted that RNs experienced a need to adapt and remould their professional role when working with PCC. Remoulding the professional role was an iterative process in which they had to challenge their

thinking and practice, which could generate insecurity;[69] despite being a challenging process, the RNs experienced this remoulding as expanding their professional role.[69] HCPs experiencing development of the professional role when working with PCC was also described in the study by Allerby *et al*.[67] In this systematic review, uncertainty towards the new professional role when providing PCC emerged due to the new practice and routines, and an experienced decrease in control and autonomy.[50 51 55 57] Having a system with a PCC culture with well-supported and committed HCPs is a prerequisite for implementing PCC.[70] Moreover, Moore *et al*[26] described how HCPs must adopt a new professional role when working with PCC. HCPs' integration of PCC is a process that requires time for reflecting on and adapting to the theory and practice of PCC for the specific context.[14] A more equal distribution of power, with patients being more involved and seen as experts in their own care, could be experienced as challenging for HCPs used to traditional care. Summer Meranius *et al*[71] argued that PCC could lead to reduced autonomy and negative health impacts for HCPs. To avoid negative health impacts, overcome the challenges of adapting, and instead expand the professional role, HCPs should be provided with time for the process and be encouraged to reflect on the theory and practice of PCC in their specific setting.

HCPs unable to provide the care desired from an ethical standpoint were also apparent. Similar to the finding of Juthberg and Sundin,[72] guilt and inadequacy could arise from being torn between striving to provide PCC, task-oriented care and organisational structures.[51 54 55 57 62 63 65] This was also described in the study by Petersson *et al*,[68] where care structures requiring tasks being carried out in high tempo could inhibit PCC. A review by Güney *et al*[32] showed that the organisation and task-oriented care could be barriers to PCC and that improved support could modify organisational barriers. Ethical stress can also emerge from not being able to provide the care desired due to a lack of resources.[27] Furthermore, perceived stress and job dissatisfaction can increase healthcare turnover rates.[7] However, systematic organisational support when providing PCC can reduce stress of conscience,[73] indicating that systematic organisational support could serve as a preventive measure. Providing resources for HCPs working with PCC could thus potentially reduce stress of conscience and improve job satisfaction; in the long term, this might lead to reduced turnover, with HCPs being more satisfied with the care provided.

HCPs also experienced improved job satisfaction and meaningfulness when providing PCC, being engaged and spending time with patients,[51–54 56 58 61 65 66] and they described reduced workload and stress from working with PCC.[51 54 66] The mixed experiences found in this review regarding engagement, workload and stress align with Summer Meranius *et al*,[71] suggesting that engagement in patient care is essential for PCC and enhances the relationship between HCPs and patients; however, too much engagement and commitment can also be considered a risk.[71] The contrasting views of workload and stress may be mediated by organisational support. Whereas reduced stress seemed to be related to resources such as enhanced support, increased stress was related to a lack of resources in terms of space, time and staffing. Structures promoting collaboration with colleagues and more time with patients facilitate a conducive working environment, promoting PCC.[70] This underpins the benefits of supporting HCPs with sufficient resources to provide PCC. Moreover, it aligns with WHO's efforts to promote the co-creation of care in the interest of improved working conditions.[74]

The HCPs felt increased appreciation from colleagues when providing PCC.[53 62 64 66] Additionally, improved team collaboration when providing PCC was described as leading to enhanced satisfaction with the work environment.[53 61 62 66] The findings regarding those themes are well aligned with Montgomery *et al*,[75] who found a positive association between well-functioning teamwork and engagement. Similarly, Allerby *et al*[67] found that PCC increased satisfaction for HCPs from the improved work environment, communication and team collaboration. This suggests that collaboration in providing PCC could have a role in improving the health and well-being of HCPs.

As in the review by van den Pol-Grevelink,[31] increased thriving, motivation and commitment were also apparent.[52 57 59 64 66] Keyko *et al*[76] found that HCPs engaged in their work will experience increased motivation and well-being, resulting in higher-quality care. Moreover, healthy HCPs are a prerequisite for the consistent provision of PCC.[77] HCPs increasingly valued their personal well-being, which meant that they could provide better, more satisfying care.[64] HCPs providing care that helps them thrive and be satisfied could mean healthier HCPs and decreased turnover in healthcare, facilitating the continuous provision of PCC.

## Strengths and limitations

A strength of this review was that its authors had different professional backgrounds, with KG, AF and MA coming from nursing, CvD from sociology and MB and GH from social medicine. In the selection and research process, pre-understanding was addressed through regular discussions in the group. Moreover, a librarian was consulted, several databases were searched, a broad range of terms was used and the reference lists of the included studies were scrutinised. Also, two authors independently screened the titles, abstracts and full texts for inclusion.

A systematic review was a suitable method for the present research purpose, since the aim was to explore and synthesise findings regarding a particular research question.[78] To demonstrate transparency, reduce the risk of bias and enable replicability, a PRISMA protocol was published in PROSPERO before data collection, a PRISMA flow diagram was used and data from extracted studies were presented in tables.[46 79–81] It was a strength that the synthesis strictly followed the steps described by Thomas and Harden,[37] since these have been proven

viable for thematically synthesising qualitative experiences of healthcare implementations. The themes derived from the qualitative synthesis can be valuable as guidance for further research or can inform policy and practice concerning HCPs' job satisfaction.[78 82]

Some potentially relevant studies might not have been captured since only English-language studies, specific search terms and studies published in or since 2010 were included. Recent studies are more likely to capture the current healthcare situation. Additionally, the literature search was performed on 21 December 2021, and studies published after that date were not included in the synthesis of this review. However, an updated search was conducted in March 2023 to capture relevant studies published after the initial literature search.

The search was delimited regarding the countries included in the eligibility criteria. Two studies were included despite being partly conducted in countries not included in the criteria, which could be considered a limitation. Although Vassbø *et al*[66] collected some of the data in Australia, two of the three countries represented in the data collection (ie, Sweden and Norway) fit the inclusion criteria and the study contained rich data, so the study was included after discussions. The study by Pinkert *et al*[65] was conducted in Germany and Austria, with the latter not being included in the criteria; however, these two countries have similar healthcare systems and the working conditions in the dementia care and acute settings are similar, so this study was also deemed relevant in order to facilitate a comprehensive overview.[65 83] Additionally, the included studies applied different methodologies and theoretical approaches within qualitative research, which might have affected how the studies were interpreted and compared.

## CONCLUSION

This qualitative systematic review identified HCPs experiences of job satisfaction when providing PCC in European healthcare settings. Noteworthy are the experiences of a new professional role, with its own demands and initial disorientation and uncertainty. Vitally, the new role came with experiences of job satisfaction related to meaningfulness, deepened relationship between HCPs and patients, appreciation from patients and colleagues and enhanced team collaboration. The findings are in line with theoretical assumptions of PCC not mainly aiming at job satisfaction, but improved relations from partnership and meaningfulness. An implication of our findings is the possibility for healthcare organisations to improve the initial phase of implementing PCC. This could be achieved by facilitating structures promoting team collaboration, and resources such as time, space and staffing. Future studies with a longitudinal follow-up approach with recurrent data collection can contribute with a deeper understanding of the mechanisms involved in the process towards increased job satisfaction when PCC is implemented. It can also contribute with causal inferences not possible from qualitative or cross-sectional studies.

**Author affiliations**
[1]Centre for Person-Centred Care (GPCC), University of Gothenburg, Gothenburg, Sweden
[2]Erasmus School of Health Policy & Management, Erasmus University Rotterdam, Rotterdam, The Netherlands
[3]Institute of Health and Care Sciences, Sahlgrenska Academy, University of Gothenburg, Gothenburg, Sweden
[4]Region Västra Götaland, Research, Education, Development and Innovation, Primary Health Care, Gothenburg, Sweden
[5]Department of Care Science, Faculty of Health and Society, Malmö University, Malmö, Sweden
[6]School of Public Health and Community Medicine, Institute of Medicine, Sahlgrenska Academy, University of Gothenburg, Gothenburg, Sweden

**Acknowledgements** We wish to thank Linda Hammarbäck at the Biomedical Library for contributing to the search strategy.

**Contributors** All study authors; KG, CvD, AF, MA, MB, and GH, were involved in planning and developing the design of the study. KG created the eligibility criteria and search strategy, which all authors provided feedback on in discussions. KG developed the protocol and screening tool, which was reviewed by AF and GH who provided comments and suggestions for improvements. KG and CvD screened titles, abstracts and full-text studies for inclusion, which was discussed with AF in a meeting. Extraction and synthesis of the data was conducted by KG, supported by GH through discussions and then reviewed by all authors providing comments and suggestions for revisions. Both KG and GH conducted the quality assessments. KG drafted the first version of the manuscript with constructive input from CvD, AF, MA, MB and GH. Throughout the research process, the manuscript and its analyses, interpretations and conclusions were continuously discussed and critically assessed by all authors. The final version of the manuscript was approved by all authors. GH is the guarantor of the study.

**Funding** This work was supported by grants from the AFA Insurance (no.190030), an organisation owned by Sweden's labour market parties, and the University of Gothenburg Centre for Person-Centred Care (GPCC), Sweden. GPCC is funded by the Swedish Government's grant for Strategic Research Areas, Care Sciences (Application to Swedish Research Council no. 2009–1088). The funders have no role in the design of the study, data collection, analysis or interpretation.

**Competing interests** None declared.

**Patient and public involvement** Patients and/or the public were not involved in the design, or conduct, or reporting, or dissemination plans of this research.

**Patient consent for publication** Not applicable.

**Ethics approval** Not applicable.

**Provenance and peer review** Not commissioned; externally peer reviewed.

**Data availability statement** Data sharing is not applicable as no data sets generated and/or analysed for this study. This is a systematic review, and all the data used are taken from previously published material.

**ORCID iDs**
Kristoffer Gustavsson http://orcid.org/0000-0002-1743-9576

Cornelia van Diepen http://orcid.org/0000-0001-6991-9443
Andreas Fors http://orcid.org/0000-0001-8980-0538

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
