## [Reviewer comments · BMJ Open]

ARTICLE DETAILS

TITLE (PROVISIONAL)	Healthcare professionals' experiences of job satisfaction when providing person-centred care: a systematic review of qualitative studies
AUTHORS	Gustavsson, Kristoffer; van Diepen, Cornelia; Fors, Andreas; Axelsson, Malin; Bertilsson, Monica; Hensing, Gunnel

VERSION 1 – REVIEW

REVIEWER	Fekonja, Urška University Medical Centre Maribor
REVIEW RETURNED	19-Feb-2023

GENERAL COMMENTS	- Very detailed and distinctly described »Methods« although I suggest to clearly define time frame of systematic research (beginning-end).- Please highlight what new can this study brings to an international audience.- Why were the quantitative studies considered as exclusion criteria?- Please ensure that your literature review is comprehensive and up to date.- Conclusions- this is a section that readers skip to, after all this work and analysis there must be stronger conclusions to present to the reader.- Chapter Strengths and limitations: Are there any limitations related to time frame of the research?- In title, there is mentioned that systematic review is made of qualitative studies from Europe, but in the article there is included one from Australia. Think about adapting title of article to the content of analysis. Just suggestion.-Another suggestion: a table of inclusion and exclusion criteria would enrich the article content.P10L54: What is »GP practices«? Define.P18L31: Please describe and explain this sentence in more detail, it is not clear what patient behaviour had an impact on satisfaction. What kind of difficult behaviour exactly is meant?P18L59: List some examples of HCPs standards and expectations regarding ethics and routines.Table 1: Which one of three types of Grounded theory was used in included study of Coyne, 2015 in Ireland? Same question in study of Uittenbroek et al, 2018 in The Netherland?
---

REVIEWER	Laserna Jiménez, Cristina University of Barcelona
REVIEW RETURNED	12-Mar-2023

GENERAL COMMENTS	Comments to the Author: Thank you for the opportunity to review this interesting manuscript. This qualitative review addresses the topic of healthcare professionals' experiences on job satisfaction providing person-centred care in healthcare settings in Europe. This issue is critical, given its relation on patient healthcare and well-being. Overall comments: The research strengths include the qualitative systematic review search performed through several databases. Also, the findings are shown through a concise summary and well- presented data and the discussion section address all the elucidated themes. The research weakness concerns the quality assessment of the included studies exclusively. P8: Selection process: In this section you should provide the number of studies you removed to be duplicated, the abstracts or titles you screened, and the exclusion criteria of the full-text studies you assessed and not included. You should mention in this section the PRISMA flow diagram (Figure 1) (Page et al., 2020) and not in the Results section. P8: Quality assessment: Despite of I can't review the tool that authors used to assess the quality of the studies because it is available only in Swedish, I wonder about why did you perform the quality appraisal of the included studies only by an author? At PROSPERO register you indicated it will be performed by two independent authors. Please, this section requires to identify how the tool you used rate. What are the maximum and minimum scores that studies can obtain and the meaning of these maximum/minimum scores? You should describe the tool in detail. P20 L11-17: Discussion section: This section doesn't need to repeat that studies are published between 2010 and 2021. This paragraph is shown as results and not as discussion. It appears as redundant and not provides discussion with the scientific evidence available. P3 L17 and P6 L55: CINHAL (in capital letters) P4 L31: Why didn't you use as a keyword the words "systematic review"? I hope this review serve you to improve your manuscript.
--

VERSION 1 – AUTHOR RESPONSE

Reviewer comments	Changes in manuscript
Reviewer 1	
Very detailed and distinctly described »Methods« although I suggest to clearly define time frame of systematic research (beginning-end).	Thank you. The following is now described and added in the manuscript to present the systematic search more clearly: Page 5, line 149-150: “A systematic search was executed on 21 December 2021 in the electronic databases CINAHL, Pubmed, and Scopus, covering the main research areas relevant to this review.” Page 7, line 187: “The selection of studies was conducted between 25 January 2022 until 11 March 2022.” Additionally, the following is added on page 6, line 150-152: “An updated search, using the same search strategy for studies published in the period between January 2021 – March 2023 was conducted in March 2023.”
Please highlight what new can this study brings to an international audience.	Excellent suggestion. The following is now added to the introduction, page 5, line 126-130: “PCC implementation being increasingly integrated into healthcare systems internationally will expose more HCPs to this model of care. Findings from this systematic review can help enhance the understanding of how HCPs’ job satisfaction relates to PCC, and lead to further research and new relevant policies to improve working conditions in healthcare.”
Why were the quantitative studies considered as exclusion criteria?	Thank you very much for this relevant question. A scoping review of quantitative studies on the topic has already been conducted by the same research group (Van Diepen et al., 2020). Therefore, the group discussed and concluded that capturing also the qualitative

	experiences of HCPs on this topic appears to be lacking and something that can contribute to filling a research gap.
Please ensure that your literature review is comprehensive and up to date.	An updated search has now been conducted, which yielded two additional relevant studies. The identified studies from the updated search will not be included as studies part of the systematic review. However, it is now stated in the text that they were identified in the updated search and that the studies will be discussed in the discussion section in relation to the review findings from the synthesis. Page 6, line 150-152: “An updated search, using the same search strategy for studies published in the period between January 2021 – March 2023 was conducted in March 2023.” Page 8, line 206-209: “For the updated literature search conducted in March 2023, the same electronic databases and search strategies were used, but the year of publication was limited to 2021-2023. The records were imported into Endnote and screened for relevance by A1. Two relevant studies were identified, which will be incorporated in the discussion section.” Page 16, line 69-72: “In the updated literature search, two additional relevant qualitative studies by Allerby et al,⁶⁷ and Petersson et al⁶⁸ were identified. Findings from these studies were in line with findings from this systematic review and will be incorporated into the following discussion.” Page 16, line 76-80: “Remoulding the professional role was an iterative process in which they had to challenge their thinking and practice, which could generate insecurity,⁶⁹ despite being a challenging process, the RNs experienced this remoulding as expanding their professional role.⁶⁹ HCPs experiencing development of the professional role when working with PCC was also described in the study by Allerby et al.⁶⁷”

	Pages 16-17, line 93-97: “HCPs unable to provide the care desired from an ethical standpoint were also apparent. Similar to the finding of Juthberg and Sundin,⁷² guilt and inadequacy could arise from being torn between striving to provide PCC, task-oriented care, and organisational structures.^{50 53 54 56 61 62 64} This was also described in the study by Peterson et al,⁶⁸ where care structures requiring tasks being carried out in high tempo could inhibit PCC.”
Conclusions- this is a section that readers skip to, after all this work and analysis there must be stronger conclusions to present to the reader.	Thank you. We agree and have now elaborated on that part to hopefully strengthen the conclusions. Please see page 19, whole conclusion paragraph: Conclusion “This qualitative systematic review identified HCP experiences of job satisfaction when providing PCC in healthcare European health care settings. Noteworthy is the experiences of a new professional role, with its own demands and initially disorientation and uncertainty. Vitally, the new role came with experiences of job satisfaction related to meaningfulness, deepened relationship between HCPs and patients, appreciation from patients and colleagues, and enhanced team collaboration. The findings are in line with theoretical assumptions of PCC not mainly aiming at job satisfaction, but improved relations from partnership and meaningfulness. An implication of our findings are the possibility for healthcare organisations to improve the initial phase of implementing PCC. This could be achieved by facilitating structures promoting team collaboration, and resources such as time, space and staffing. Future studies with a longitudinal follow-up approach with recurrent data collection can contribute with a deeper understanding of the mechanisms involved in the process towards increased job satisfaction when PCC is implemented. It can also contribute with causal inferences not possible from qualitative or cross-sectional studies.”

	Changes in abstract conclusion page 2, line 50-55: “Conclusion: This systematic review found varied experiences among HCPs. Notably, the new professional role was experienced to entail disorientation and uncertainty; importantly, it also entailed experiences of job satisfaction such as meaningfulness, an improved relationship between HCPs and patients, appreciation, and collaboration. To facilitate PCC implementation, healthcare organisations should focus on supporting HCPs through collaborational structures, and resources such as time, space and staffing.”
Chapter Strengths and limitations: Are there any limitations related to time frame of the research?	Thank you for this comment. The following is now added in strengths and limitations section page 18, line 155-158: “Additionally, the literature search was performed on 21 December 2021, and studies published after that date were not included in the synthesis of this review. However, an updated search was conducted in March 2023 to capture relevant studies published after the initial literature search.” If the comment refers to whether there is a limitation with the study only including studies during or after 2010, please see page 18, line 153-155: “Some potentially relevant studies might not have been captured since only English-language studies, specific search terms, and studies published in or since 2010 were included. Recent studies are more likely to capture the current healthcare situation.”

In title, there is mentioned that systematic review is made of qualitative studies from Europe, but in the article there is included one from Australia. Think about adapting title of article to the content of analysis. Just suggestion.	Thank you for this suggestion. The words “from Europe” is now removed from the title. New title: Healthcare professionals’ experiences of job satisfaction when providing person-centred care: a systematic review of qualitative studies
Another suggestion: a table of inclusion and exclusion criteria would enrich the article content.	Great suggestion. We agree that this improves the study, and this is now arranged. Please see Table 1 on page 6. The associated number of the other tables have been changed accordingly.
P10L54: What is »GP practices«? Define.	Page 10, line 276, the following was added to clarify: (an organisation of one or more General Practitioners).
P18L31: Please describe and explain this sentence in more detail, it is not clear what patient behaviour had an impact on satisfaction. What kind of difficult behaviour exactly is meant?	We agree with this, but unfortunately the original article does not provide any details regarding what difficult behaviours they are referring to apart from that PCC did not influence their job satisfaction if the patients did not respond positively. However, the sentence is now rephrased. Page 14, line 16-18: “Some HCPs found that PCC did not influence their job satisfaction, especially when dealing with difficult behaviours of patients, such as when they did not respond well.”
P18L59: List some examples of HCPs standards and expectations regarding ethics and routines.	This is an excellent suggestion, and examples are now added. Pages 14-15, line 30-33: “PCC provision was aligned with HCPs’ standards and expectations regarding ethics and routines, such as providing equitable and inclusive care, discussing goals and writing health plans with patients.”
Table 1: Which one of three types of Grounded theory was used in included study of Coyne, 2015 in Ireland? Same question in study of Uittenbroek et al, 2018 in The Netherland?	Thanks for pointing that out. Information on this is now added in Table 2 for Coyne, Uittenbroek, and Öhman.

Reviewer 2	
Comments to the Author: Thank you for the opportunity to review this interesting manuscript. This qualitative review addresses the topic of healthcare professionals' experiences on job satisfaction providing person-centred care in healthcare settings in Europe. This issue is critical, given its relation on patient healthcare and well-being.	
Overall comments: The research strengths include the qualitative systematic review search performed through several databases. Also, the findings are shown through a concise summary and well- presented data and the discussion section address all the elucidated themes. The research weakness concerns the quality assessment of the included studies exclusively.	Thank you very much for these comments. We agree concerning the quality assessment, which has now been conducted by one more author. Our revisions concerning this are provided further down in this table related to the comment about the quality assessment, so please see there for more information.
P8: Selection process: In this section you should provide the number of studies you removed to be duplicated, the abstracts or titles you screened, and the exclusion criteria of the full-text studies you assessed and not included. You should mention in this section the PRISMA flow diagram (Figure 1) (Page et al., 2020) and not in the Results section.	Thank you. Search and selection results subheading is removed. The requested information is now inserted in the methods section under 'Selection process' on page 7-8. Selection process "The selection of studies was conducted between 25 January 2022 until 11 March 2022. The 3754 records from the database searches were imported into Endnote for reference management and into Rayyan software for screening. The first (A1) and second author (A2) screened titles and abstracts blinded and independently for relevance in Rayyan using a screening tool (see Online Supplementary File 3).⁴³ The 'detect duplicates' option available in both programs was used to remove 872 duplicates.^{44 45} The remaining 2882 studies were screened for title and abstract, and 2807 of those were excluded. Out of the 75 studies, one was not retrieved in full-text. In the second step of the selection process, the remaining 74 studies were downloaded and shared in an Endnote library. The studies were read in full text for relevance against the inclusion criteria by A1 and A2 independently in Endnote. All full-text studies were discussed in a meeting among A1, A2, and A3 until consensus was reached. Fifty six studies were excluded due to incorrect outcome, exposure, or design, or by being conducted in the wrong location, or not

	being written in English. Eighteen studies were considered relevant to the review. After closer reading, one study was interpreted as related to working during the COVID-19 pandemic rather than to person-centred work; therefore, it was excluded at this stage after discussions. The selection process is presented in a PRISMA flow diagram (Figure 1).⁴⁶ Figure 1. The selection of studies presented in a PRISMA flow diagram.⁴⁶ For the updated literature search conducted in March 2023, the same electronic databases and search strategy were used, but the year of publication was limited to 2021-2023. The records were imported into Endnote and screened for relevance by A1. Two relevant studies were identified, which will be incorporated in the discussion section.”
P8: Quality assessment: Despite of I can't review the tool that authors used to assess the quality of the studies because it is available only in Swedish, I wonder about why did you perform the quality appraisal of the included studies only by an author? At PROSPERO register you indicated it will be performed by two independent authors. Please, this section requires to identify how the tool you used rate. What are the maximum and minimum scores that studies can obtain and the meaning of these maximum/minimum scores? You should describe the tool in detail.	Thank you. We agree that the assessment should be conducted by two independent reviewers as planned in the PROSPERO register. In addition to A1, also A6 has now conducted the quality assessment, which was in agreement with A1's quality assessments. However, limitation descriptions of two studies were discussed and revised to become more accurate (Please see Online Supplementary File 5, and the limitation descriptions for Barbosa et al and Ross et al). The following is added under the quality assessment subheading on pages 8-9, line 216-232: The studies were initially assessed by A1, and secondly independently assessed by A6, who is a senior researcher and Professor. Both A1 and A6 considered omitting studies indicating low methodological quality. All studies considered relevant were assessed as having moderate or high quality by both assessors. Items in the five domains were related to methodological choices, such as if the aim was coherent with regard to the theoretical stance, if the recruitment was suitable and well conducted, or if the method of analysis was appropriate and carried out properly. The items could be answered with “yes”, “no” or “unclear”. The combined item answers regarding whether it had serious methodological concerns affecting the reliability of the study resulted in a domain grading of “yes” “no” or “unclear” (see Online Supplementary File 4). The gradings from the

	five domains were then combined and given an overall rating of the study as follows: no or few concerns = high quality; moderate concerns = moderate quality; and high concerns = low quality, in line with the recommendations of SBU.⁴⁹ The studies' ratings varied between moderate ($n = 12$) and high ($n = 5$) quality. The most common deficiencies were missing descriptions of the researcher's role and pre-understanding, and how it might have affected the study results. No studies were assessed as having low methodological quality, so no studies were excluded due to the quality assessment. The quality assessments are presented in Online Supplementary File 5.
P20 L11-17: Discussion section: This section doesn't need to repeat that studies are published between 2010 and 2021. This paragraph is shown as results and not as discussion. It appears as redundant and not provides discussion with the scientific evidence available.	We agree. The following is now removed from first paragraph in the discussion: “Seventeen studies published between 2010 and 2021 were synthesised. Most of them were conducted in Sweden, followed by the UK. Most participants in the included studies were women. Participants were mainly RNs or specialist nurses, closely followed by assistant nurses, students, enrolled nurses, care workers, and frontline staff. The themes identified in this review present a pattern of both positive and negative experiences of PCC in relation to job satisfaction.”
P3 L17 and P6 L55: CINHAL (in capital letters)	Changed.
P4 L31: Why didn't you use as a keyword the words “systematic review”?	Thank you for this suggestion. It is now added.

Apart from the changes carried out in accordance with reviewers' comments, the following changes have been conducted:

- One additional systematic review on HCP outcomes and PCC was found (Brownie, S., & Nancarrow, S. (2013). Effects of person-centered care on residents and staff in aged-care facilities: a systematic review. *Clinical interventions in aging*, 8, 1–10. <https://doi.org/10.2147/CIA.S38589>) and added into the introduction on page 5, line 133-137:

“Five reviews of aspects of HCP outcomes and PCC have been identified,³⁰⁻³⁵ which presented mixed findings. These reviews focused on specific professions, such as registered nurses (RNs), nurse aides, direct-care workers, and caregivers in residential care or nursing homes,^{30-32 34 35} or included only quantitative studies.^{30 31 33-35}”

- Small adjustments to the arrows and boxes position in the PRISMA flow diagram.

VERSION 2 – REVIEW

REVIEWER	Fekonja, Urška University Medical Centre Maribor
REVIEW RETURNED	09-May-2023

GENERAL COMMENTS	Thank you for opportunity to review this interesting systematic review. In my opinion, I think that revision of manuscript is well done and ready for publication. It was my pleasure to review this manuscript.
---

REVIEWER	Laserna Jiménez, Cristina University of Barcelona
REVIEW RETURNED	17-Apr-2023

GENERAL COMMENTS	Comments to the Author: The authors addressed all the comments made contributing to the improvement of the first manuscript. The weaknesses presented by the study at the beginning of this assessment were resolved. I would suggest to the authors only a concern: L206: Why did you perform an updated search between January 2021 to March 2023 if the first search was conducted in December 2021?
--

VERSION 2 – AUTHOR RESPONSE

Reviewer 1:

Comment: Thank you for opportunity to review this interesting systematic review. In my opinion, I think that revision of manuscript is well done and ready for publication.

It was my pleasure to review this manuscript.

Answer: Thank you for your comment, we are glad that you find that our changes have been conducted satisfactorily.

Reviewer 2:

Comment: “L206: Why did you perform an updated search between January 2021 to March 2023 if the first search was conducted in December 2021?”

Answer: Thank you very much for noticing this mistake. It has now been changed, please see page 6, line 150-153 “An updated search, using the same search strategy for studies published in the period between December 2021 – March 2023 was conducted in March 2023.”

Additional changes not related to reviewer comments:

-In the first sentence of the conclusion, the word “healthcare” was written twice on page 19, line 175. This has now been changed.

-In Online Supplementary file 5, the reference numbers were not in accordance with the main document. A new supplementary file has been added where the references match the main document.